# Short- and Long-Term Outcomes of Left Main Coronary Artery Stenting in Patients Disqualified from Coronary Artery Bypass Graft Surgery

**DOI:** 10.3390/jpm12030348

**Published:** 2022-02-25

**Authors:** Wojciech Jan Skorupski, Marta Kałużna-Oleksy, Maciej Lesiak, Aleksander Araszkiewicz, Włodzimierz Skorupski, Stefan Grajek, Przemysław Mitkowski, Małgorzata Pyda, Marek Grygier

**Affiliations:** 1st Department of Cardiology, Poznań University of Medical Sciences, 61-848 Poznań, Poland; marta.kaluzna@wp.pl (M.K.-O.); maciej.lesiak@skpp.edu.pl (M.L.); aaraszkiewicz@interia.pl (A.A.); w.skorupski@wp.pl (W.S.); stefan.grajek@skpp.edu.pl (S.G.); przemyslaw.mitkowski@skpp.edu.pl (P.M.); malgorzata.pyda@skpp.edu.pl (M.P.); mgrygier@wp.pl (M.G.)

**Keywords:** coronary artery disease, LM stenting, multivessel disease, heart team, PCI

## Abstract

The study aims to evaluate the short- and long-term outcomes of left main percutaneous coronary interventions (LM PCI) in patients disqualified from coronary artery bypass graft surgery (CABG). We included 459 patients (mean age: 68.4 ± 9.4 years, 24.4% females), with at least 1-year follow-up; 396 patients in whom PCI was offered as an alternative to CABG (Group 1); and 63 patients who were disqualified from CABG by the Heart Team (Group 2). The SYNTAX score (29.1 ± 9.5 vs. 23.2 ± 9.7; *p* < 0.001) and Euroscore II value (2.72 ± 2.01 vs. 2.15 ± 2.16; *p* = 0.007) were significantly higher and ejection fraction was significantly lower (46% vs. 51.4%; *p* < 0.001) in Group 2. Patients in Group 2 more often required complex stenting techniques (33.3% vs. 16.2%; *p* = 0.001). The procedure success rates were very high and did not differ between groups (100% vs. 99.2%; *p* = 0.882). We observed no difference in periprocedural complication rates (12.7% vs. 7.8%; *p* = 0.198), but the long-term all-cause mortality rate was higher in Group 2 (26% vs. 21%; *p* = 0.031). LM PCI in patients disqualified from CABG is an effective and safe procedure with a low in-hospital complication rate. Long-term results are satisfactory.

## 1. Introduction

The left main coronary artery (LM) provides over 75% of the blood flow to the left ventricular (LV) myocardium, even in patients with dominant right coronary artery (RCA) [1]. Therefore, a significant LM stenosis puts a patient at high risk since it can endanger nearly the entire LV myocardium. Furthermore, LM stenosis characterizes the worst prognosis of all forms of coronary artery disease (CAD). LM stenosis is detected in about 4–9% of patients undergoing coronary angiography [2,3].

Earlier studies showed that the 3-year mortality rate in patients with unprotected left main coronary artery disease (LM-CAD) receiving only medical therapy was nearly 50% [4,5,6]. Coronary artery bypass graft surgery (CABG) improves survival rates compared with medical therapy. It has been the standard of care for the LM-CAD for nearly four decades [7,8,9]. However, advances in percutaneous coronary interventions (PCI) lead to the reconsideration of the PCI role in the LM-CAD treatment [10,11]. Contemporary PCI standards for LM-CAD include pre-procedural imaging, functional assessment, procedural planning using new-generation drug-eluting stents (DES), lesion preparation, proximal optimization technique, kissing balloon technique, post-procedural imaging, and potent dual antiplatelet therapy [12].

Current European Society of Cardiology (ESC) guidelines still favor CABG as the only treatment method for LM disease with diffuse CAD [13]. However, in less advanced CAD, LM PCI is a method of choice. The safety and efficacy of PCI in comparison with CABG in patients with unprotected LM-CAD were proved in randomized trials with the use of first-generation and second-generation DESs [14,15,16,17]. Present guidelines apply only to patients with coronary anatomy suitable for both procedures and with a low risk of death during surgery [13]. Thus, many patients do not meet the guidelines criteria. Based on current recommendations, the Heart Team should select the optimal treatment strategy, considering the patient’s individual characteristics [13]. However, clinical status and severe comorbidities in many patients do not allow CABG performance. In these patients, LM PCI is the only alternative and should be considered the best treatment choice. Moreover, it should be underlined that patients disqualified from CABG are generally excluded from clinical trials; thus, the real results of the procedure and long-term follow-up data are not well known in this population. Therefore, personalized patient-to-patient decisions are crucial. Additional information would be extremely useful in supporting the clinical decision-making process and developing future medical progress towards better LM-CAD disease treatment.

We aimed to evaluate the short- and long-term outcomes of LM PCI in patients disqualified from CABG surgery in a real-world setting.

## 2. Materials and Methods

### 2.1. Study Population

From January 2015 to June 2019, 613 consecutive patients underwent LM PCI in our department. Subsequent 459 patients, with at least 1-year follow-up, were included in a prospective registry presented in this paper (Figure 1). 

We included patients with ≥50% diameter stenosis of LM with or without the involvement of left anterior descending artery (LAD), ostial circumflex coronary artery (LCx), or both. In patients with moderate lesions, intravascular ultrasound imaging (IVUS) was used to confirm the significance of the lesion, with a cut-off value of minimal lumen area of 6.0 mm^2^ for LM. Patients with LM equivalent disease, i.e., distal bifurcation Medina 0.1.1, who presented <70% stenoses of the ostial LAD or LCx without any evidence of ischemia in its myocardial distribution, were not included in the study [16].

The study group consisted of 396 patients in whom PCI was offered as an alternative to CABG (Group 1) and 63 patients who were disqualified from CABG (Group 2). Group 1 consisted of patients eligible for both PCI and CABG. In Group 2, all patients had clear angiographic indications to CABG, but were disqualified from the procedure by two experienced cardiac surgeons because of prohibitive perioperative risk due to serious cardiac or extra-cardiac comorbidities (Table 1).

### 2.2. Study Procedures

After a Heart Team meeting with a cardiac surgeon, invasive procedures were performed by experienced invasive cardiologists at a high-volume referral center with cardiac surgery department on-site. Patients were treated with the intention to achieve complete revascularization of all major vessels with significant lesions. All bifurcation lesions were assessed angiographically according to the Medina classification [18]. Periprocedural myocardial infarction (type 4a) was diagnosed based on ESC Fourth Universal Definition of Myocardial Infarction (2018) [19]. The use of glycoprotein IIb and IIIa inhibitors and imaging modalities (IVUS, OCT) was at the operator’s discretion. In the analyzed population, IVUS or OCT imaging was used in 142 (30.9%) patients, which was beyond this analysis’s scope.

We analyzed the clinical and angiographic data of these patients, as well as the short- and long-term outcomes. Baseline clinical data were collected for each patient at the index procedure. Main procedural data with all periprocedural and in-hospital complications were also collected and analyzed. The follow-up data were collected by telephone contact or based on the official records of the National Health Fund. Terminal patients whose expected survival was less than one year were excluded from long-term survival analysis. Chronic kidney disease (CKD) was defined as a decreased kidney function based on glomerular filtration rate (GFR) < 60 mL/min within the last three months and was calculated by the Cockcroft–Gault formula [20]. Troponin analysis was not available in the whole analyzed population, but only in patients included in the prospective part of the study, in which troponin levels were measured routinely before and between 12- and 24-h after the procedure.

The registry conformed to the ethical guidelines of the 1975 Declaration of Helsinki and was granted ethics approval by the Institutional Review Board of the Poznan University of Medical Sciences.

### 2.3. Study Endpoints

The composite primary outcome of the study was defined as the rate of in-hospital death, in-hospital myocardial infarction, and long-term all-cause death. 

### 2.4. Statistical Analysis

Statistical analysis was performed using STATISTICA 12 (Tibco Software Inc., Palo Alto, CA, USA). A standard descriptive statistic was applied in the analysis. All continuous variables are presented as means (standard deviation) or medians (interquartile range). The normality distribution was analyzed using the Shapiro–Wilk test. The statistical significance of differences was tested with the t–Student test or nonparametric U Mann–Whitney test. Categorical variables were reported as counts or percentages and compared by tests for proportions. The Kaplan–Meier method was used to calculate the survival probability at follow-up. The survival curves were compared with a log–rank test. A two-sided *p* value of <0.05 was considered significant for all the tests.

## 3. Results

### 3.1. Study Population

The study cohort consisted of 459 consecutive patients who underwent LM PCI (mean age: 68.4 ± 9.4 years, 75.6% males). Patients baseline characteristics are presented in Table 2. Patients disqualified from CABG (Group 2) were older (70.9 ± 10.9 years vs. 68 ± 9.1 years; *p* = 0.024), with more females (34.9% vs. 22.7%; *p* = 0.036), and with higher prevalence of CKD (54% vs. 31.6%; *p* < 0.001). The groups did not differ in other cardiovascular risk factors. There were more patients with prior CABG in Group 1. This difference results from the fact that patients after CABG are qualified more often for PCI than for re-CABG. However, patients after prior CABG were not disqualified from surgery by the protocol. On admission, stable CAD was less frequent in Group 2 (46% vs. 61.4%; *p* = 0.021), and patients with NSTEMI were found more often (22.2% vs. 7.8%; *p* < 0.001). Left ventricular ejection fraction (LVEF) was significantly lower in Group 2 (46 ± 11.4% vs. 51.4 ± 11%; *p* < 0.001) with no significant differences in other echocardiographic parameters. The Euroscore II value was higher in Group 2 (2.72 ± 2.01 vs. 2.15 ± 2.16; *p* = 0.007). 

### 3.2. CAD Characteristics

The extent of coronary artery disease is shown in Table 3. The groups did not differ within the localization of the LM disease. Protected LM was more common in Group 1. The lack of right coronary artery support (i.e., recessive, critical stenosis, or occlusion of RCA) was more common in Group 2. The presence of non-LM lesions did not differ significantly in LAD and RCA. Still, it was found more frequently in non-ostial LCx in Group 2. The incidence of two- or three-vessel disease was also higher in Group 2. The frequency of Medina 1.1.1 bifurcation was significantly higher in Group 2, as well as the mean values of the Syntax score (29.1 ± 9.5 vs. 23.2 ± 9.7; *p* < 0.001).

### 3.3. LM PCI Procedure

The PCI procedure characteristics are shown in Table 4. An early success rate was very high (99%) and did not differ between the groups. The number (1.9 ± 0.9 vs. 1.63 ± 0.79; *p* = 0.012) as well as the total length (43.7 ± 22.3 vs. 37.1 ± 21.2; *p* = 0.009) of implanted stents were significantly higher in Group 2. Periprocedural fluoroscopy time and a dose of radiation did not differ significantly. Artery access was similar in both groups, and it was more often a radial approach. All LM lesions were stented with second-generation DESs. Various stenting techniques were used. Two-stent techniques were more common in Group 2 (33.3% vs. 16.2%; *p* = 0.001). Among the two-stent techniques, the crush technique was the most common (48.2%). All LM procedures were carried out without mechanical LV support.

### 3.4. Clinical Outcomes

Troponin elevation after PCI was relatively high in both groups (57.1% vs. 46.7%; *p* = 0.136) (Table 5). All periprocedural complications, including stroke, tamponade, artery dissection, and contrast-induced nephropathy (12.7% vs. 7.8%; *p* = 0.198), as well as periprocedural mortality and myocardial infarction type 4a (7.9% vs. 4%; *p* = 0.294), did not differ between the groups. 

The median follow-up was 808 days (min: 366 days, max: 1616 days, interquartile range: 606 days). At long-term follow-up, a higher all-cause mortality rate was observed in Group 2 (26% vs. 21%; *p* = 0.031) (Figure 2).

## 4. Discussion

To the best of our knowledge, based on the literature review, the current study is the first to report the long-term outcomes of patients with LM-CAD disqualified from CABG in a real-world setting. 

The risk models, used in current ESC guidelines and based on SYNTAX score, rely entirely on anatomical characteristics, and define three categories of risk (low, intermediate, and high) based on conventional thresholds (≤22, 23–32, and ≥33, respectively) [1]. Current guidelines indicate that PCI is an appropriate alternative to CABG in LM disease with low anatomical complexity of CAD (class Ia of recommendation) and should be considered for PCI in LM with intermediate CAD (class IIa of recommendation) [13]. In patients with LM-CAD and high SYNTAX score, the guidelines suggest better survival after CABG, so PCI should not be undertaken in that subgroup of patients (PCI class III of recommendation) [13]. However, the real risk is imprecise due to the lack of these patients in large randomized clinical trials. Therefore, the personalized patient-to-patient Heart Team decision with detailed analysis of the individual clinical status and cardiac and non-cardiac comorbidities is crucial here. 

Our study provides information about real-life LM PCI and includes patients who should have been qualified for CABG but for some reasons were not eligible for the surgery. This population is normally excluded from clinical trials, and the available information about this group of patients is insufficient. The principal finding from the present study is that PCI of LM in patients who were disqualified from CABG for various cardiovascular and non-cardiovascular reasons was a safe procedure with similar complication rates as in patients who had the same recommendation class to PCI or CABG. Our results showed that patients, who underwent LM PCI in a high-volume center, also those disqualified from CABG, had favorable short- and long-term (median: 808 days) outcomes. If we compared the group of patients qualified originally for PCI (Group 1) and those disqualified from CABG (Group 2) treated with LM PCI, there were no differences between the groups in periprocedural complications rates or occurrence of in-hospital cardiovascular events and death. These perioperative values obtained in our study are similar to those presented in the NOBLE or EXCEL studies. The periprocedural myocardial infarction after PCI occurred in 5% in the NOBLE study, 3.6% in the EXCEL study, and 4.6% in our study. The in-hospital death rate was <1% in the NOBLE study, and 0.4% in our study [17,21]. Our short-term results are also consistent with outcomes of other registries [22,23,24]. 

In the long-term follow-up, with the median of 808 days, significantly higher mortality in Group 2 than Group 1 was noted (26% vs. 21%; *p* = 0.031). Considering the characteristics of the disqualified group, this is understandable, and the results obtained still demonstrate a favorable prognosis. Additionally, our long-term mortality outcomes do not diverge from the results of other real-world investigations [25], and we cannot simply compare these results with the results of selected, large, randomized trials such as PRECOMBAT, EXCEL, or NOBLE [15,16,17]. It should be emphasized that the real-world results presented in our study showed that the frequency of comorbidities was somehow higher than in selected RCTs populations [16,17]. In patients disqualified from CABG, the CKD rate was especially high (54%) and significantly higher than in Group 1, where the CKD rate was 31.6% (*p* < 0.001). This does not correspond with the results from the EXCEL trial [16], where in the group treated with PCI, renal insufficiency (defined as GFR < 60 mL/min) occurred only in 17.7% [16]. In addition, in our analysis, patients in Group 2 had NSTEMI at admission more often (22.2%) than those from the EXCEL study, where that percentage was only 13.2% [16]. Our observation showed similar outcomes, despite the worse clinical status of patients resulting mainly from the large NSTEMI rate and a greater percentage of comorbidities affecting prognosis. The survival rates of patients in Group 1 in our study, with a 21.0% mortality rate after 5 years, agree with large LM registries, i.e., LE MANS or MAIN-COMPARE Registry [22,26].

PCI of LM-CAD is a complex procedure that requires procedural planning based on clinical and anatomical characteristics with an individual choice of stenting technique [12,27]. In our investigation, as in the EXCEL study, only second-generation DESs were used [16]. The stent strategy is generally based on the burden of atherosclerotic disease at the level of the LCx ostium. The diameter stenosis of ≥70%, lesion length of >10 mm, and/or a difficult side branch access indicate a complex lesion that typically requires upfront a two-stents technique. In all other cases, a provisional approach is preferable [28]. In our study, the two-stent technique was more often used in the group disqualified from CABG because these patients presented with more complex bifurcation lesions. Moreover, the rate of bifurcations Medina 1.1.1 was also higher (*p* = 0.016). There is a widespread controversy in the literature regarding choosing a one- or two-stent technique for the treatment of LM bifurcation lesions. From time to time, the experts’ consensus tries to refine the data and establish up-to-date recommendations [29]. Non-randomized data uniformly suggests that outcomes are worse with a two-stent strategy [30,31,32], but randomized data support the double kissing (DK) crush technique for a true LM bifurcation [33] and prefers it over culotte [34]. A recent randomized controlled study showed that among patients with true LM bifurcation stenosis requiring intervention, fewer major adverse cardiac events occurred with a stepwise provisional approach than with planned dual stenting. However, the difference was not statistically significant [35].

Additionally, in Group 2, atherosclerotic lesions requiring revascularization in other parts of coronary arteries, especially in LCx (*p* = 0.041), and LM with the 2-vessel disease were observed more frequently (*p* = 0.002). Consequently, the total number of implanted stents, as well as the total length of stents, were higher, which can indirectly affect the outcomes [36]. It should be emphasized that despite the higher total stent length in Group 2 in our study, the numbers in both groups still did not exceed the length obtained in the other studies with second-generation DESs [27]. The complexity of the procedure was higher in Group 2 also due to the lack of support from RCA, which occurred more often in those patients (*p* = 0.002). Despite using more complex techniques in Group 2, no significant increase in troponin levels or myocardial infarction type 4a rates were observed if compared with the other group. Most patients analyzed in our study had distal LM bifurcation or trifurcation disease with involvement of at least one side branch (MEDINA 1.1.1, 1.0.1, 1.1.0), which was similar to the population from the EXCEL study and the recently published BBK-LM Registry [37,38]. 

All findings indicate that patients in Group 2 had more advanced CAD and, following current guidelines, should be the candidates for CABG [13]. Our results showed the safety of PCI procedure with a very high success rate also in these patients, giving them a chance for better outcomes compared with conservative treatment. The comparison with previous trials should be made with caution due to the lack of information of patients with a high SYNTAX score (≥33) undergoing LM PCI. The EXCEL study analyzed only patients with a low and intermediate risk assessed at site according to the SYNTAX score [16]. However, after evaluation by the core laboratory, it was found that there was a group of patients with a high SYNTAX score (≥33) [37]. Therefore, this study confirmed the safety and effectiveness of LM PCI in patients with a high SYNTAX score, who were also presented in our study [37]. In the NOBLE study, in a small group of 46 patients with a high SYNTAX score, the MACCE rate (death from any cause, non-procedural myocardial infarction, repeat revascularization, or stroke) was of over 30% [17]. The long-term results of our study presented a slightly better prognosis in high SYNTAX score group with a mortality of approximately 25%.

The presented study is an analysis of a real-world cohort of patients. In contrast to other huge trials, it described patients with LM disease disqualified from the CABG procedure, who are mainly excluded from clinical trials. However, our analysis has several limitations. Firstly, the presented study was a prospective registry, but not all clinical data were available. Secondly, troponin analysis was not available in the entire analyzed population. Thirdly, this was a real-world study, and LM disease was not a homogenous disease. Outcomes could have been affected by the location of the disease (ostial/shaft/bifurcation), the complexity of the lesion, and distal CAD. Finally, the presented study analyzed in-hospital outcomes, as well as the long-term outcomes; however, the long-term follow-up showed only all-cause mortality rates.

## 5. Conclusions

LM PCI in patients disqualified from CABG is an effective and safe procedure with low in-hospital complication rates. Long-term results in this group of patients are satisfactory. To provide an individualized approach, the data should be considered by physicians and patients together when deciding on the revascularization strategy because this life-saving treatment remains the only option for such patients.

## Figures and Tables

**Figure 1 jpm-12-00348-f001:**
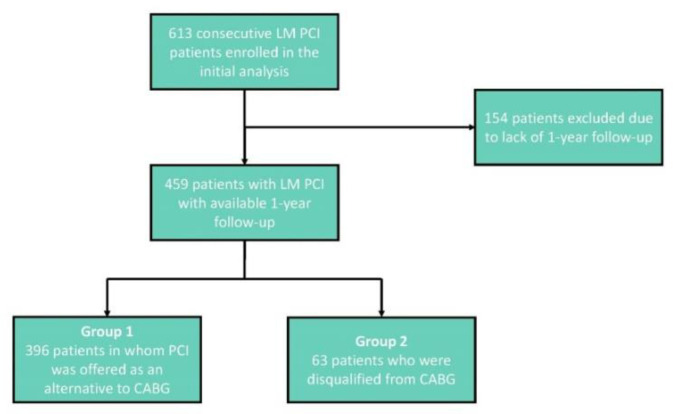
Flowchart presenting the inclusion and exclusion criteria. LM—Left Main, PCI—Percutaneous Coronary Intervention, CABG—Coronary Artery Bypass Graft.

**Figure 2 jpm-12-00348-f002:**
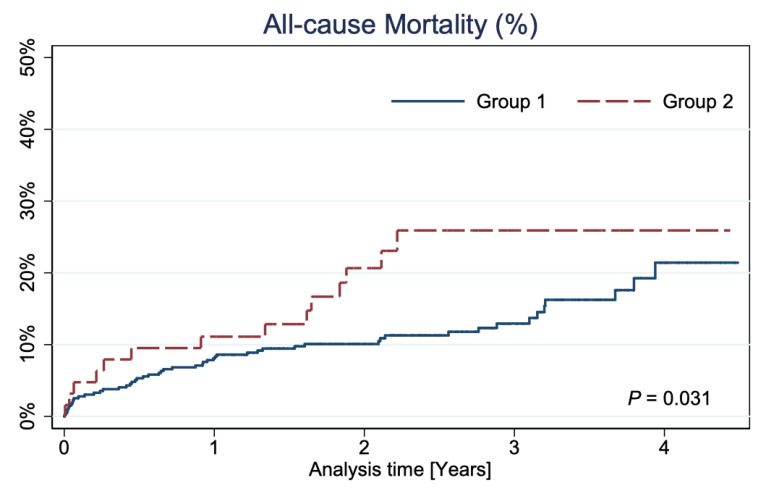
Kaplan–Meier curves showing all-cause mortality.

**Table 1 jpm-12-00348-t001:** Clinical conditions leading to CABG disqualification in the study group.

Disqualification Factors			
Cardiac Factors	Number of Patients	Extra-Cardiac Factors	Number of Patients
Enlargement of LV (LVEDD > 70 mm)	6	Severe obesity (BMI > 35 kg/m^2^)	19
LVEF < 30%	4	Renal failure (GFR < 30 mL/min)	13
Diffusely diseased peripheral segments (no useful for CABG)	3	Multilevel peripheral atherosclerosis	10
Valve diseases—not eligible for surgery	2	Frailty syndrome	9
		Advanced COPD (GOLD ≥ 3)	8
		Mental disorders	7
		Others *	23

LV—Left Ventricle, LVEDD—Left Ventricular End Diastolic Diameter, LVEF—Left Ventricle Ejection Fraction, CABG—Coronary Artery Bypass Graft, BMI—Body Mass Index, GFR—Glomerular Filtration Rate, COPD—Chronic Obstructive Pulmonary Disease, GOLD—The Global Initiative for Chronic Obstructive Lung Disease; * Advanced oncology disease, musculoskeletal disorders, age over 85 years, carotid arteries stenosis, active ulcerative colitis.

**Table 2 jpm-12-00348-t002:** Study population baseline characteristics.

Variable	Total *n* = 459	Group 1 *n* = 396	Group 2 *n* = 63	*p*-Value (Group 1 vs. Group 2)
Age (y)	68.4 ± 9.4	68 ± 9.1	70.9 ± 10.9	0.024
Gender (female)	112 (24.4%)	90 (22.7%)	22 (34.9%)	0.036
BMI (kg/m^2^)	28.1 ± 4.6	28.1 ± 4.4	28.2 ± 5.8	0.353
Hypertension	375 (81.7%)	326 (82.3%)	49 (77.8%)	0.386
Hyperlipidemia	230 (50.1%)	203 (51.2%)	27 (42.9%)	0.215
CKD	159 (34.6%)	125 (31.6%)	34 (54%)	<0.001
DM	164 (35.7%)	136 (34.3%)	28 (44.4%)	0.120
Stroke/TIA	34 (7.4%)	29 (7.3%)	5 (7.9%)	0.863
COPD	37 (8.1%)	29 (7.3%)	8 (12.7%)	0.145
PVD	66 (14.4%)	56 (14.1%)	10 (15.9%)	0.716
AF	59 (12.9%)	48 (12.1%)	11 (17.5%)	0.240
Smoking (current)	168 (36.6%)	140 (35.4%)	28 (44.4%)	0.164
Prior MI	227 (49.5%)	199 (50.3%)	28 (44.4%)	0.382
Stable CAD	272 (59.2%)	243 (61.4%)	29 (46%)	0.021
Unstable CAD	125 (27.2%)	108 (27.3%)	17 (27%)	0.962
NSTEMI	45 (9.8%)	31 (7.8%)	14 (22.2%)	<0.001
STEMI	13 (2.8%)	10 (2.5%)	3 (4.8%)	0.558
Prior PCI LAD	104 (22.7%)	91 (23%)	13 (20.6%)	0.672
Prior PCI LCX	65 (14.2%)	59 (14.9%)	6 (9.5%)	0.253
Prior PCI RCA	132 (28.8%)	118 (29.8%)	14 (22.2%)	0.217
Prior CABG	92 (20%)	88 (22.2%)	4 (6.3%)	0.003
LVEDD (mm)	51.6 ± 7.7	51.5 ± 7.6	52.4 ± 8.2	0.523
LVEF (%)	50.6 ± 11.2	51.4 ± 11	46 ± 11.4	<0.001
EuroScore II	2.32 ± 2.13	2.15 ± 2.16	2.72 ± 2.01	0.007
Syntax Score	24.0 ± 9.9	23.2 ± 9.7	29.1 ± 9.5	<0.001
0–22 (low)	230 (50.1%)	214 (54%)	16 (25.4%)	<0.001
23–32 (intermediate)	145 (31.6%)	120 (30.3%)	25 (39.7%)	0.137
≥33 (high)	84 (18.3%)	62 (15.7%)	22 (34.9%)	<0.001

BMI—Body Mass Index, CKD—Chronic Kidney Disease, DM—Diabetes Mellitus, TIA—Transient Ischemic Attack, COPD—Chronic Obstructive Pulmonary Disease, PVD—Peripheral Vascular Disease, AF—Atrial Fibrillation, MI—Myocardial Infarction, CAD—Coronary Artery Disease, PCI—Percutaneous Coronary Intervention, LAD—Left Anterior Descending Artery, LCx—Left Circumflex, RCA—Right Coronary Artery, CABG—Coronary Artery Bypass Graft, LVEDD—Left Ventricular End Diastolic Diameter, LVEF—Left Ventricle Ejection Fraction.

**Table 3 jpm-12-00348-t003:** Coronary artery disease characteristics.

Variable	Total *n* = 459	Group 1 *n* = 396	Group 2 *n* = 63	*p*-Value (Group 1 vs. Group 2 )
LM distal	375 (81.7%)	319 (80.1%)	56 (88.9%)	0.087
LM bifurcation	292 (63.6%)	249 (62.9%)	43 (68.3%)	0.410
LM trifurcation	52 (11.3%)	43 (10.9%)	9 (14.3%)	0.425
LM calcification	70 (15.3%)	54 (13.6%)	16 (25.4%)	0.016
LAD disease (not ostial)	240 (52.3%)	202 (51%)	38 (60.3%)	0.167
LCx disease (not ostial)	159 (34.6%)	130 (32.8%)	29 (46%)	0.041
Protected LM	63 (13.7%)	62 (15.7%)	1 (1.6%)	0.003
RCA recessive (a)	29 (6.3%)	23 (5.8%)	6 (9.5%)	0.260
RCA with critical stenosis (b)	70 (15.3%)	54 (13.6%)	16 (25.4%)	0.016
RCA total occlusion (c)	82 (17.9%)	68 (17.2%)	14 (22.2%)	0.331
Lack of RCA support to LM-CAD (a + b + c)	156 (34%)	124 (31.3%)	32 (50.8%)	0.002
Extent of diseased vessels				
LM only	126 (27.5%)	118 (29.8%)	8 (12.7%)	0.005
LM plus 1-vessel disease	164 (35.7%)	144 (36.4%)	20 (31.7%)	0.477
LM plus 2-vessel disease	116 (25.3%)	90 (22.7%)	26 (41.3%)	0.002
LM plus 3-vessel disease	53 (11.5%)	44 (11.1%)	9 (14.3%)	0.464
Bifurcation Medina				
1.0.0	94 (20.5%)	83 (21%)	11 (17.5%)	0.522
1.0.1	37 (8.1%)	32 (8.1%)	5 (7.9%)	0.968
1.1.0	91 (19.8%)	80 (20.2%)	11 (17.5%)	0.612
1.1.1	70 (15.3%)	54 (13.6%)	16 (25.4%)	0.016

LM—Left Main, LAD—Left Anterior Descending Artery, LCx—Left Circumflex Artery, RCA—Right Coronary Artery, LM-CAD—Left Main Coronary Artery Disease.

**Table 4 jpm-12-00348-t004:** Left Main PCI procedure characteristics.

Variable	Total *n* = 459	Group 1 *n* = 396	Group 2 *n* = 63	*p*-Value (Group 1 vs. Group 2)
PCI success	456 (99.3%)	393 (99.2%)	63 (100%)	0.882
Number of stents	1.67 ± 0.81	1.63 ± 0.79	1.9 ± 0.9	0.012
Total length of implanted stents (mm)	38.0 ± 21.5	37.1 ± 21.2	43.7 ± 22.3	0.009
Fluoroscopy time (min)	17.47 ± 9.25	17.16 ± 9.17	19.42 ± 9.57	0.060
Radiation dose (mGy)	1442 ± 877	1427 ± 879	1531 ± 871	0.370
Contrast volume (mL)	247.4 ± 94.2	248.1 ± 96.9	242.9 ± 76.3	0.804
Arterial Access site				
Radial	270 (58.8%)	235 (59.3%)	35 (55.6%)	0.570
Femoral	189 (41.2%)	161 (40.7%)	28 (44.4%)	
Stenting LM only	57 (12.4%)	50 (12.6%)	7 (11.1%)	0.735
Stenting LM bifurcation				
One-stent technique	317 (69.1%)	282 (71.2%)	35 (55.5%)	0.013
Two-stents technique	85 (18.5%)	64 (16.2%)	21 (33.3%)	0.001
Two-stents techniques	Total *n* = 85	*n* = 64	*n* = 21	
Crush	30 (35.3%)	18 (28.1%)	12 (57.1%)	0.016
DK-Crush	11 (12.9%)	9 (14.1%)	2 (9.5%)	0.879
Cullote	1 (1.1%)	0 (0%)	1 (4.8%)	0.247
T-stenting	19 (22.4%)	15 23.4%)	4 (19%)	0.905
Provisional stenting	24 (28.2%)	22 (34.3%)	2 (9.5%)	0.028

PCI—Percutaneous Coronary Intervention, LM—Left Main, DK-Crush—Double Kissing Crush Technique.

**Table 5 jpm-12-00348-t005:** Periprocedural outcomes.

Variable	Total *n* = 459	Group 1 *n* = 396	Group 2 *n* = 63	*p*-Value (Group 1 vs. Group 2)
Significant troponin elevation (5×) after PCI	222 (48.4%)	185 (46.7%)	36 (57.1%)	0.136
Myocardial Infarction	21 (4.6%)	16 (4%)	5 (7.9%)	0.294
In-hospital Death	2 (0.4%)	2 (0.5%)	0 (0%)	0.642
Stroke	1 (0.2%)	0 (0%)	1 (1.6%)	0.137
Tamponade	2 (0.4%)	2 (0.5%)	0 (0%)	0.642
Pulmonary oedema	1 (0.2%)	0 (0%)	1 (1.6%)	0.137
Dissection of aorta	1 (0.2%)	1 (0.3%)	0 (0%)	0.291
Perforation of femoral artery	1 (0.2%)	1 (0.3%)	0 (0%)	0.291
Contrast induced nephropathy	17 (3.7%)	12 (3%)	5 (7.9%)	0.120

PCI—Percutaneous Coronary Intervention.

## Data Availability

The data presented in this study are available on request from the corresponding author.

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
