# Peer review of "Short- and Long-Term Outcomes of Left Main Coronary Artery Stenting in Patients Disqualified from Coronary Artery Bypass Graft Surgery"

_jpm, 2022, doi:10.3390/jpm12030348_

Round 1

Reviewer 1 Report

In this manuscript, authors report the outcome of left main coronary artery stenting in patients who are disqualified for CABG and ultimately underwent with PCI. The content, description specially of the discussion part and overall presentation of this real-world study are praiseworthy. Although the study population size is small to conclude anything absolutely and similar results have already been obtained by many randomized trials which the authors also agree with. The limitations that the authors indicate also could be ventured for further studies. Ultimately, the last line of the conclusion really conclude the scenario of current curable possibilities. I have some minor concerns about presentation. 

  1. On line 70, LCx has been defined as "ostial circumflex coronary arteries" while on line 146, 159 etc it has been abbreviated as Left circumflex. As the later one is more familiar for the audience it could be similar for all the places.
  2. Table 2 should be accommodate in a single page. 
  3. A flow chart could be included to present the population size, inclusion and exclusion criteria for easier understanding. 
  4. It would be interesting to know the long term status of those "disqualification factors" specially the cardiac after stenting and how far it impacted on that "all-cause mortality" graph.  

Author Response

Comment 1: On line 70, LCx has been defined as "ostial circumflex coronary arteries" while on line 146, 159 etc it has been abbreviated as Left circumflex. As the later one is more familiar for the audience it could be similar for all the places.

Response: We sincerely thank the reviewer for this comment. On line 70, the sentence, “We included patients with ≥ 50% diameter stenosis of LM with or without the involvement of left anterior descending artery (LAD), ostial circumflex coronary artery (LCx), or both.” defines the study inclusion criteria and applies to ostial LCx. However, on line 159, the sentence, “The presence of non-LM lesions did not differ significantly in LAD and RCA. Still, it was found more frequently in LCx in Group 2.” concerns the issue of the frequency of not ostial LCx lesions in both groups.

We have corrected the sentence for clarity:

“Still, it was found more frequently in not ostial LCx in Group 2.”

Comment 2: Table 2 should be accommodate in a single page. 

Response: We appreciate the reviewer’s comment. We followed the recommendations.  During the submissions process in the system, the tables move slightly beyond our control. Of course, we will make sure that in the final version of the manuscript, the table is on single page.

Comment 3: A flow chart could be included to present the population size, inclusion and exclusion criteria for easier understanding. 

Response: We followed the recommendations. The flowchart has been added.

Comment 4: It would be interesting to know the long term status of those "disqualification factors" specially the cardiac after stenting and how far it impacted on that "all-cause mortality" graph.

Response: We appreciate the reviewer’s comments. Because some patients had more than one disqualification factor, the long-term survival analysis and Kaplan-Meier charts of individual cardiac and extra-cardiac disqualification factors and their impact on all-cause mortality are not suitable for inclusion in the manuscript due to the variety of combinations and relatively small number of patients in each subgroup.

Reviewer 2 Report

The article is devoted to a very popular question nowadays - what is better for a patient with CAD: PCI or CABG? As for conservative therapy with multiple lesions of the coronary arteries, it is more or less certain - direct complete revascularization. The second-generation stent market appearance has led in all developed countries to a decrease in the number of CABG (on pump and off pump) and an increase in the use of PCI. Therefore, the task of the authors of the article is clear and relevant.

There are some disadvantages for the reader:

  1. PCI technique is not specified in patients with diffuse CAD with unprotected LM-CAD.
  2. What percent made it possible to open the RCA with stenting? Did it lead to complete revascularization?
  3. Unfortunately, the authors do not provide data on long-term survival, which is more than 5 years of conventionally operated patients with diffuse CAD for comparison with PCI at their center.

Author Response

Comment 1: PCI technique is not specified in patients with diffuse CAD with unprotected LM-CAD.

Response: We appreciate the reviewer’s comment. The incidence of two- or three-vessel disease was higher in Group 2. Various stenting techniques were used. The choice of strategy was at the operator’s discretion. Two-stent techniques were more common in Group 2 (33.3% vs. 16.2%; p=0.001). Among the two-stent techniques, the Crush and DK-crush techniques were the most common (48.2%). Provisional stenting technique was used in 28.2% of patients.

The following sentences are included in the "Results" section:

“The incidence of two- or three-vessel disease was also higher in Group 2.”

“Two-stent techniques were more common in Group 2 (33.3% vs. 16.2%; p=0.001). Among the two-stent techniques, the crush technique was the most common (48.2%).”  

Comment 2: What percent made it possible to open the RCA with stenting? Did it lead to complete revascularization?

Response: Revascularization of CTO was performed in approximetly 10% of RCA CTO patients. Complex analysis of the PCI RCA outcomes after LM procedure was beyond the scope of this study and total revascularization was not the primary endpoint. The decision regarding CTO-RCA treatment was taken after PCI of LM. So far, there is no clear scientific evidence that the revascularization of  CTO improves patients survival.

Comment 3: Unfortunately, the authors do not provide data on long-term survival, which is more than 5 years of conventionally operated patients with diffuse CAD for comparison with PCI at their center.

Response: We aimed to evaluate the short- and long-term outcomes of LM PCI in patients disqualified from CABG surgery in a real-world setting. In Group 2, all patients had clear angiographic indications to CABG but were disqualified from the procedure by two experienced cardiac surgeons because of prohibitive perioperative risk due to serious cardiac or extra-cardiac comorbidities.

Present guidelines apply only to patients with coronary anatomy suitable for both procedures and with a low predicted surgical mortality. Therefore, it is impossible to create a group of patients treated with CABG for comparison with Group 2 from our study.

Reviewer 3 Report

I would like to thank the authors on this prospective registry. This is a very important topic: addressing the outcome of PCI in patients who are rejected by surgeons. and doing it in a registry form gives us an idea about real-life practice. I have few comments and clarifications: 1-Abstract: Lines 18-20: the authors state that Syntax and Euroscore were higher in group 2 and the EF was lower, but the sequence of numbers should be changed. Since they mentioned Group 1 first, readers would understand that the first value is for Group 1 and the second value is for Group 2, because in the previous sentence, the authors defined what the groups were in this order: group 1 then group 2. 2-Materials and methods: -The authors had 613 patients but enrolled 459 patients only. Why was that? was it because of exclusion criteria, or patients were missed (perhaps due to death)? - The authors did not state how long they intended to follow the patients and how long is long-term follow up. 3-Results: - Table 2 (syntax score categories), table 3 (extent of diseased vessels) and (bifurcation medina), Table 4 (stenting techniques) should all have one p value for the list of categories, because the comparison should be between all the categories and not between each category and the rest. -Table 4: why did the authors differentiate between Crush and DK-Crush? both are crush techniques, and they commented on them in the text as a single item. Table 4: Success rate in Group 1 was 99%, what happened to the 3 failed cases? 4-Discussion: -The authors say that the patients disqualified from CABG had favorable outcome, but they should state clearly that they did poorly compared with Group 1 on long term. They also state that they may be a better option than medical treatment, but in fact the rate of all cause mortality was 26% and we do not know what would be the outcome of medical treatment in the current era with advances in medical therapy which are far better than the older medical management on which older studies in the 1980s ( medical ttt vs CABG) were based. -Line 285: the authors state the long term results of the high syntax group was better than in NOBLE study. But, they did not mention how this subgroup did in this registry and what were their outcomes.

Author Response

Comment 1: Abstract: Lines 18-20: the authors state that Syntax and Euroscore were higher in group 2 and the EF was lower, but the sequence of numbers should be changed. Since they mentioned Group 1 first, readers would understand that the first value is for Group 1 and the second value is for Group 2, because in the previous sentence, the authors defined what the groups were in this order: group 1 then group 2.

Response: We appreciate the reviewer’s comment. Both in the manuscript and the abstract, we use the method in which the first group in the text is the test group, i.e. (Group 2 vs. Group 1; p value). We believe that this is a generally accepted worldwide method. Of course, if the Editor and Reviewers express the need to do so, we will change the order of the numbers throughout the text.

Comment 2: Materials and methods: -The authors had 613 patients but enrolled 459 patients only. Why was that? was it because of exclusion criteria, or patients were missed (perhaps due to death)? - The authors did not state how long they intended to follow the patients and how long is long-term follow up.

Response: From January 2015 to June 2019, 613 consecutive patients underwent LM PCI in our department. Subsequent 459 patients, with at least 1-year follow-up, were included in a prospective registry presented in this paper. A flowchart presenting population size, inclusion and exclusion criteria has been added. The median follow-up was 808 days (min: 366 days, max: 1616 days, interquartile range: 606 days).

The following sentences are included in the "Results" section:

“The median follow-up was 808 days (min: 366 days, max: 1616 days, interquartile range: 606 days).”

Comment 3: Results: - Table 2 (syntax score categories), table 3 (extent of diseased vessels) and (bifurcation medina), Table 4 (stenting techniques) should all have one p value for the list of categories, because the comparison should be between all the categories and not between each category and the rest.

Response: Our goal from the beginning was to compare individual categories between groups, tests for the proportions were performed.

In the case of comparing individual fractions between two groups, our method is appropriate. The proportion test is the same as the chi-square test in the case of 2x2 tables (two categories into two categories), while in the case of more than two categories, we can combine the other categories that will complement the analyzed fraction. The result is the significance of the selected fraction between the groups (Group 1 vs Group 2) in relation to the entire sample, and not the analysis of the differences of the two categories in the two samples. We realize that such a procedure would not be possible in the case of tables e.g. 3x3. Then, one of the variables treated as a grouping variable would lead to a comparison of more than 2 groups simultaneously, which would require the use of post-hoc analysis.

Comment 4: Table 4: why did the authors differentiate between Crush and DK-Crush? both are crush techniques, and they commented on them in the text as a single item.

Response: Crush and DK-crush are two techniques with different results [1] that belong to a “family” of crush techniques. We believe that separating them in the table makes educational and informational sense.

[1] Chen, S.L., Zhang, J.J., Ye, F., Chen, Y.D., Patel, T., Kawajiri, K., Lee, M., Kwan, T.W., Mintz, G. and Tan, H.C. (2008), Study comparing the double kissing (DK) crush with classical crush for the treatment of coronary bifurcation lesions: the DKCRUSH-1 Bifurcation Study with drug-eluting stents. European Journal of Clinical Investigation, 38: 361-371. https://doi.org/10.1111/j.1365-2362.2008.01949.x

Comment 5: Table 4: Success rate in Group 1 was 99%, what happened to the 3 failed cases?

Response: Case 1: The patient developed cardiogenic shock after qualification for PCI LM. The procedure was performed before initiating the Impella program in our cath lab. Despite the use of the IABP patient died. Case 2: The catheter caused dissection of ostial LM propagating to the ascending aorta. Pericardial tamponade and cardiopulmonary arrest in the PEA mechanism occurred. Case 3: After trying to cross through subtotal heavily calcified LM patient's condition deteriorated, haemodynamic instability began to develop, temporary loss of consciousness occurred, and we had to stop the procedure.

Comment 6: Discussion: -The authors say that the patients disqualified from CABG had favorable outcome, but they should state clearly that they did poorly compared with Group 1 on long term.

Response: We appreciate the reviewer’s comment.

The following sentences are included in the "Discussion" section:

“In the long-term follow-up, with the median of 808 days, significantly higher mortality in Group 2 than Group 1 was noted (26% vs. 21%; p=0.031). Considering the characteristics of the disqualified group, this is understandable, and the results obtained still demonstrate a favorable prognosis.”

Comment 7: They also state that they may be a better option than medical treatment, but in fact the rate of all cause mortality was 26% and we do not know what would be the outcome of medical treatment in the current era with advances in medical therapy which are far better than the older medical management on which older studies in the 1980s ( medical ttt vs CABG) were based.

Response: Earlier studies showed that the 3-year mortality rate in patients with unprotected left main coronary artery disease (LM-CAD) receiving only medical therapy was nearly 50% [2] According to 2018 ESC/EACTS Guidelines on myocardial revascularization treatment of LM disease with stenosis > 50% is mandatory. The 3-year mortality rate in LM CAD patients receiving only medical therapy from early studies was nearly 50% and was almost twice as high as in patients in Group 2 in our study.

[2] Conley MJ, Ely RL, Kisslo J, Lee KL, McNeer JF, Rosati RA. The prognostic spectrum of left main stenosis. Circulation 1978, 57, 947–52.

Comment 8: -Line 285: the authors state the long term results of the high syntax group was better than in NOBLE study. But, they did not mention how this subgroup did in this registry and what were their outcomes.

Response: We sincerely thank the reviewer for this comment.

We have added more information to the manuscript for clarification:

“In the NOBLE study in a small group of 46 patients with a high SYNTAX score the MACCE rate (death from any cause, non-procedural myocardial infarction, repeat revascularization, or stroke) was of over 30% [17]. The long-term results of our study presented a slightly better prognosis in high SYNTAX score group with a mortality of approximately 25%.”